# Genome-Wide Identification of Banana *Csl* Gene Family and Their Different Responses to Low Temperature between Chilling-Sensitive and Tolerant Cultivars

**DOI:** 10.3390/plants10010122

**Published:** 2021-01-08

**Authors:** Weina Yuan, Jing Liu, Tomáš Takáč, Houbin Chen, Xiaoquan Li, Jian Meng, Yehuan Tan, Tong Ning, Zhenting He, Ganjun Yi, Chunxiang Xu

**Affiliations:** 1Department of Pomology, College of Horticulture, South China Agricultural University, Guangzhou 510642, China; weina_yuan@stu.scau.edu.cn (W.Y.); liujing@stu.scau.edu.cn (J.L.); hbchen@scau.edu.cn (H.C.); mengjian@stu.scau.edu.cn (J.M.); yehuantan@stu.scau.edu.cn (Y.T.); ningtong@stu.scau.edu.cn (T.N.); hezhenting@stu.scau.edu.cn (Z.H.); 2Centre of the Region Haná for Biotechnological and Agricultural Research, Czech Advanced Technology and Research Institute, Palacký University Olomouc, 783 75 Olomouc, Czech Republic; tomas.takac@upol.cz; 3Institute of Biotechnology, Guangxi Academy of Agricultural Sciences, Nanning 530007, China; lixiaoquan@gxaas.net; 4Institute of Fruit Tree Research, Guangdong Academy of Agricultural Sciences, Guangzhou 510640, China

**Keywords:** banana (*Musa* spp.), cellulose synthase-like genes, genome-wide identification, hemicellulose, low temperature stress

## Abstract

The cell wall plays an important role in responses to various stresses. The cellulose synthase-like gene (*Csl*) family has been reported to be involved in the biosynthesis of the hemicellulose backbone. However, little information is available on their involvement in plant tolerance to low-temperature (LT) stress. In this study, a total of 42 *Csls* were identified in *Musa acuminata* and clustered into six subfamilies (*CslA*, *CslC*, *CslD*, *CslE*, *CslG*, and *CslH*) according to phylogenetic relationships. The genomic features of *MaCsl* genes were characterized to identify gene structures, conserved motifs and the distribution among chromosomes. A phylogenetic tree was constructed to show the diversity in these genes. Different changes in hemicellulose content between chilling-tolerant and chilling-sensitive banana cultivars under LT were observed, suggesting that certain types of hemicellulose are involved in LT stress tolerance in banana. Thus, the expression patterns of *MaCsl* genes in both cultivars after LT treatment were investigated by RNA sequencing (RNA-Seq) technique followed by quantitative real-time PCR (qPCR) validation. The results indicated that *MaCslA4/12*, *MaCslD4* and *MaCslE2* are promising candidates determining the chilling tolerance of banana. Our results provide the first genome-wide characterization of the *MaCsls* in banana, and open the door for further functional studies.

## 1. Introduction

The plant cell wall consists of polysaccharides (cellulose, hemicellulose and pectin), proteins and other compounds. It plays critical roles in the maintenance of cell integrity, and the regulation of many developmental processes in plants [1,2,3,4,5,6]. The cell wall represents not only a mechanical barrier, but also a signaling component during plant responses to various biotic [7,8] and abiotic stresses [9,10].

Hemicelluloses are a diverse group of complex, non-cellulosic polysaccharides, which constitute approximately one-third of the plant cell wall. The backbones of hemicellulosic polysaccharides in plants are made of the *cellulose synthase-like* (*Csl*), which are members of a much larger superfamily of genes referred to as glycosyltransferase 2 [11,12]. Early studies of cellulose synthase (*CesA*) homologs in model plant organisms established that there are nine *Csl* families: *CslA*, *CslB*, *CslC*, *CslD*, *CslE*, *CslF*, *CslG*, *CslH* and *CslJ* [11,13,14]. Recent research in other flowering plants has identified a new *CslM* family [15].

It has been reported that *CslAs* are involved in the biosynthesis of mannan and glucomannan backbones [5,16,17,18,19], while *CslCs* are related to the synthesis of xyloglucan backbone [20,21,22]. *CslD* genes may participate in either cellulose or mannan synthesis in tip-growing cells [23,24,25,26,27], as well as xylan and homogalacturonan [28]. *CslF*, *CslH* and *CslJ* subfamilies are responsible for the synthesis of (1,3; 1,4)-β-glucan, also known as mixed-linkage glucan (MLG) synthases [29,30,31,32,33]. However, the functions of *CslB/E/G/M* remain poorly characterized [34].

Plant *Csl* genes play substantial roles in developmental processes, such as root hair formation [6], the control of organ size [4], tiller number [3] and the maintenance of adherent mucilage structure [1,2,5]. *Csl* genes were also reported to be involved in plant resistance/tolerance to biotic or abiotic stresses, such as salt [35,36], boron (B) [37,38] or heavy metal [36,39] stress, as well as pathogen infection [32,40,41]. The responses of *Csl* genes or hemicellulose components to low temperatures (LTs) have been reported in many plant species, including banana [42,43,44,45,46]. Recently, some hemicellulose metabolism-related genes were proven to play important roles in plant tolerance to LT stress [47,48]. However, the role of *Csl* genes in plant tolerance to LT stress has not been reported.

A detailed characterization of the plant *Csl* genes will be helpful for better understanding their functional and biochemical properties. To date, the whole *Csl* gene repertoire has been cataloged in rice (*Oryza sativa*) [13], poplar (*Populus trichocarpa*) [49,50], moss (*Physcomitrella patens*) [51], maize (*Zea mays*) [52,53], barley (*Hordeum vulgare*) [54], pine (*Pinus taed*) [55], tomato (*Solanum lycopersicum*) [56], pineapple (*Ananas comosus*) [57], bread wheat (*Triticum aestivum* L.) [58] and white pears (*Pyrus bretschneideri*) [59]. However, these genes have not been extensively studied in banana (*Musa* spp.), one of the most important fruit and food crops in the tropical and subtropical regions [60,61].

To screen potential *Csl* candidates that determine the chilling tolerance of banana, we provide a genome-wide characterization of *Csls* in banana (*Musa acuminata*). Moreover, changes in the hemicellulose contents in chilling-tolerant (CT) and chilling-sensitive (CS) banana cultivars after exposure to LT stress were investigated, and the expression of 42 *MaCsl* members was examined by RNA sequencing (RNA-Seq) techniques followed by validation with quantitative real-time PCR (qPCR) in the present study. These results will significantly facilitate studies focused on the functions of *MaCsls* in plant growth and development, as well as tolerance/resistance to biotic and abiotic stresses. Furthermore, we present the potential *MaCsls* candidates determining banana chilling tolerance, which might be used for the development of new banana genotypes tolerant to chilling conditions.

## 2. Results

### 2.1. The Response of Hemicellulose in Banana to LT Stress

The CS cultivar ‘Baxijiao’ (BX) exhibited an eight-times higher content of hemicellulose compared to the CT cultivar ‘Dongguandajiao’ (DJ) in the control condition (25 °C; Figure 1). However, the hemicellulose content dramatically increased in the CT cultivar grown at 16 °C, followed by a significant decrease with the further decline in temperature, but it still maintained significantly high levels. Contrary, the hemicellulose content in the CS cultivar continuously decreased with the drop in temperature. The hemicellulose content in the CS cultivar was significantly higher than that in the CT plants grown at 10 °C and 7 °C.

### 2.2. MaCsl and Their Molecular Structural Features

#### 2.2.1. Phylogenetic Analysis of the MaCsls

In this study, we constructed a multi-species phylogenetic tree of Arabidopsis (*Arabidopsis thaliana*, model species for dicots), rice (monocots), and banana (monocots) *Csl* genes based on full-length protein sequences using MEGA software. *MaCsl* subfamilies clustered together with similar *Csl* subfamilies from Arabidopsis and rice, indicating a shared evolutionary history (Figure 2). The most abundant *MaCsl* subfamilies are *CslA* and *CslC*, with 13 and 12 members, respectively. Remarkably, the *CslA* family is abundant also in Arabidopsis (9 genes) and rice (11 genes). The *MaCslD* and *MaCslE* subfamilies contain 9 and 3 genes, respectively. The *CslG* subfamily, previously reported to be specific to dicots, harbors four *Csl* members in monocotyledonous banana. Only one *CslH* member was identified in *Musa acuminata*, while the rice genome possesses three *CslHs*. In contrast to rice, which contains eight *CslFs*, this subfamily is missing in banana (as well as Arabidopsis). Both banana and rice do not have the *CslB* subfamily, the specific subfamily for dicots.

#### 2.2.2. Identification of *Csl* Genes in *Musa acuminata*

In total, we have identified 42 candidate *Csl* genes in the banana (*Musa acuminata*) genome. Based on phylogenetic relationships with Arabidopsis and rice, these 42 *MaCsls* are grouped into six subfamilies: *MaCslA*, *MaCslC*, *MaCslD*, *MaCslE*, *MaCslG*, and *MaCslH*. The 42 *MaCsl* genes are distributed over all 11 banana chromosomes. Interestingly, the *MaCslG* subfamily members are located solely on Chr8, and no other subfamily members are present on this chromosome (Figure 3, Table 1). The basic characterization of the *MaCsl* gene family, including the corresponding proteins, is shown in Table 1. The length of their open reading frames ranged from 1245 bp (*MaCslD7*) to 3657 bp (*MaCslD5*), encoding polypeptides with 415 to 1219 amino acids. The molecular weight (MW) of the polypeptides varied from 46.1 to 134.9 kD, with isoelectric points (pI) ranging from 6.4 (*MaCslD2*) to 9.5 (*MaCslD7*). The diversity in the amino acid length, MW and pI of MaCsls may indicate functional differences among the members.

#### 2.2.3. Phylogenetic Evolutionary Tree, Gene Structure, and Conserved Motifs

As shown in Figure 4A, the MaCsl proteins could be divided into four subgroups according to the phylogenetic distribution. MaCslA proteins are grouped into subgroup I, while subgroup II consists of MaCslC proteins. The MaCslD proteins belong to subgroup III, while the MaCslE, MaCslG and MaCslH proteins are present in subgroup IV.

The analysis of the exon–intron structure of the gene family can help to better understand its evolutionary trajectory. The intron/exon arrangement of 42 *MaCsls* was constructed based on the phylogenetic tree. The results showed that the exon–intron structures of the *MaCsl* genes are similar within subgroups I and II. The genes of subgroup I (*MaCslAs*) possess 7–9 introns and 8–10 exons, while only 4 introns and 5 exons are present in the genes of subgroup II (*MaCslCs*). In addition, most *MaCslA* and *MaCslC* genes possess both an upstream 5′ untranslated region (5′UTR) and a downstream 3′UTR. These results suggest a conserved evolutionary pattern of *MaCsl* genes in these two subgroups. However, a different pattern was observed in subgroups III and IV, which have a higher variation in the gene structure. For example, there are five introns in *MaCslD2*, but only two to three are found in other *MaCslDs*. Moreover, *MaCslD5* has no 5′UTR, while the other four *MaCslD* genes (*MaCslD2/3/6/7*) have neither 5′UTR nor 3′UTR (Figure 4B).

Next, the Multiple Em for Motif Elucidation (MEME) web-based application was utilized to further analyze the putative motifs of MaCsl proteins. A total of 10 conserved motifs were identified, and the relative positions of these motifs in the amino acid sequences are shown in Figure 4C. The *MaCslA* and *MaCslC* members are more closely related and contain motifs 1–7 and motif 10, while the rest, with the exception of *MaCslH*, contain motifs 8 and 9. Members with similar motif compositions can be clustered together, indicating functional similarity among the MaCsl proteins of the same subfamily.

### 2.3. Differences in the Responses of MaCsls to LT Stress between CS and CT Cultivars

In order to know the response of *MaCsls* to LT and find the *MaCsls* potentially determining banana LT stress tolerance, their expressions in CT and CS banana cultivars under different LTs were investigated using RNA-Seq techniques. The number of differentially expressed genes (DEGs) in each comparison group is shown in Appendix A. Cold responsive genes are listed in Appendix A. The different responses of the *MaCsls* to LT between the CT and CS banana cultivars are shown in Table 2. Most *MaCsls* were downregulated by LT (s). Nine *MaCsls* (*MaCslA2/6*, *MaCslC1/4*, *MaCslD3/6/7/8*, *MaCslG4*) did not respond to LTs in both cultivars. Four (*MaCslC3* and *MaCslD1/5/9*) were upregulated in both cultivars. Some other *MaCsls* were only regulated in the CT (*MaCslA4/12*, *MaCslD4*) or CS (*MaCslA8*, *MaCslC5*, *MaCslD2*) cultivar. When compared to the CS cultivar, *MaCslE2* showed a significantly higher expression level in the CT cultivar at all four tested temperature points, *MaCslC7* showed higher expression at 25 °C, 10 °C and 7 °C while *MaCslG1* at 16 °C, 10 °C, and 7 °C. Relative to the CS cultivar, the expression levels of *MaCslA10/11*, *MaCslC11* and *MaCslD4* in the CT cultivar were significantly higher at two temperature points, and the expression levels of *MaCslA13* and *MaCslG2* were higher only at 7 °C. Appendix A lists the *p*-values of differentially expressed *MaCsls*, while the fragments per kilobase of exon per million reads mapped values of 42 *MaCsls* in the two cultivars before and after LT treatments are shown in Appendix A.

To validate the results of the RNA-Seq analysis, the expressions of genes with higher expression levels under LT(s) in the CT cultivar when compared with the CS one (*MaCslA10/11/13*, *MaCslC7/11*, *MaCslD4*, *MaCslE2*, *MaCslG1/2*), and the ones induced by LTs only in the CT cultivar (*MaCslA4/12*, and overlapped *MaCslD4*), were analyzed by qPCR. As shown in Figure 5, *MaCslA4* was induced by all tested LT points in the CT cultivar, while it was upregulated only at 10 °C in the CS one. As a result, the CT cultivar showed significantly higher expression levels at 10 °C and 7 °C, though the result was opposite at 25 °C. The relative expression level of *MaCslA10* in the control CT cultivar was 151 times higher than in the CS one, and decreased dramatically under LTs. Though there was a small peak in *MaCslA10* expression in the CS cultivar at 16 °C, it was still lower than in the CT one at 10 °C and 7 °C. Similar trends were observed for *MaCslC7* and *MaCslC11*. The LT treatment resulted in an increase in *MaCslA11* expression in the CS banana, while the opposite was found in the CT cultivar, but the latter still showed a significantly higher expression level at all tested temperatures except 7 °C. *MaCslA12* was upregulated by LTs of 10 °C and 7 °C in the CT cultivar, but only 10 °C in the CS one. Furthermore, the CT banana showed higher *MaCslA12* expression at all tested LTs. *MaCslA13* was upregulated by LTs of 16 °C and 10 °C in both cultivars. *MaCslD4* was induced by LTs of 10 °C and 7 °C in both cultivars, and showed significantly higher expression levels in the CT cultivar under LTs. A decrease in expression of *MaCslE2* and *MaCslG2* was observed in LT-treated CS banana, but this was not the case for the CT one. Opposite to *MaCslC7*, the expression level of *MaCslG1* in the CS cultivar was much higher than in the CT one, and it was downregulated by LTs. Though the expression level in the CT cultivar increased at 10 °C, it was lower than that in the CS cultivar at 25 °C and 16 °C. In most cases, the qPCR confirmed the results from the RNA-Seq analysis, and contradictory results were observed only for *MaCslA13* and *MaCslG1*. In conclusion, our results suggest *MaCslD4, MaCslA4/12* and *MaCslE4* as genes with highest potential for the determination of banana chilling tolerance.

## 3. Discussion

### 3.1. The Features of MaCsls

Hemicelluloses encompass heteromannans, xyloglucan, heteroxylans, and MLG, and constitute roughly one-third of the cell wall biomass [62], and their backbones are considered to be synthesized by Csl proteins. The first report on the function of *Csl*-encoded proteins demonstrated their mannan-synthase activity in soybean (*Glycine max*) [16]. Later on, numerous evidences were reported on their capability to synthesize hemicellulose backbones [5,17,18,19,21,22,23,27,29,30,31,33,63]. However, little is known about their function in plant growth, development and stress tolerance.

A genome-wide analysis of gene family is an efficient approach for understanding gene structure, function, and evolution. To date, detailed genome-wide explorations of Csls have been limited to less than 20 plant species, such as Arabidopsis, poplar, rice, sorghum (*Sorghum bicolor*), maize, and various grasses [50,53,56,57,59,64,65]. In this study, we conducted a comprehensive analysis of the *MaCsl* gene family, including the identification of members, phylogenetic relationships, chromosomal distribution, and expression profiles in two LT-treated banana cultivars with different tolerances to LT. A total of 42 putative *Csl* genes were identified in the *Musa acuminata* genome. The number of *Csl* genes varies among the plant species, ranging from 21 in *Dendrobium catenatum* [66] to 108 in bread wheat [58]. Further, the number of *Csl* genes in these species is not proportional with the genome size, likely because the genomes of some species have undergone significant gene losses [53].

Plant *Csl* gene family could be classified into ten subfamilies (*CslA*–*CslH*, *CslJ* and *CslM*) [11,13,14,15]. Among them, *CslA*, *CslC*, and *CslD* are conserved in all land plants. The *CslB* and *CslG* families have been known to be specific for dicots [24], whereas *CslF* and *CslH* are restricted to grasses [14,67]. We have found that the genome of banana does not have the *CslF* family, but it contains *CslGs*. Similarly, *CslJ* was originally believed to be specific to grasses, but it was recently identified in some dicots [14]. On the other hand, a recent report established the presence of the *CslB* subfamily in monocots as well [68]. These results suggest that the knowledge about plant *Csl* gene family needs further examination. Banana contains only one *CslH* gene, and lacks *CslF* and *CslJ* genes, suggesting that the abundance/level of MLG in banana is much lower than in the other monocotyledonous crops, such as rice, wheat and maize, because they possess much more *CslF/H/J* genes responsible for the biosynthesis of MLG [13,53,58,69].

In bread wheat, more than half the *TaCsl* genes are concentrated on two chromosomes (chr2 and chr3 of each sub-genome) [58]. However, in banana, most chromosomes contain 4–5 *Csl* genes. This suggests a relatively even distribution of the *MaCsl* genes on these chromosomes.

### 3.2. The Involvement of MaCsls in Tolerance to LT Stress

Banana production is seriously threated by various biotic and abiotic stresses, such as Fusarium wilt and chilling stress. In the present study, the changes in hemicellulose content and the expressions of genes related to the biosynthesis of hemicellulose backbone were compared between CS and CT cultivars.

It was proposed that increased amounts of hemicelluloses were connected to enhanced cell wall stiffening, and prevented cell collapse caused by dehydration, thus contributing to plant tolerance to LTs [70,71]. In agreement, increase of hemicellulose content was observed only in the LT-treated CT cultivar in the present study, though the hemicellulose content in the CS cultivar was always higher than in the CT one, except at 16 °C. This suggests that the amount of specific types of hemicellulose, and not the total hemicellulose content, that was affecting the chilling tolerance of banana. Similarly, XXXG-rich xyloglucan, arabinoxylan and acetylated galactomannan were reported to be involved in plant desiccation tolerance [72]. The CT cultivar showed a striking increase in hemicellulose content when the temperature dropped from 25 °C to 16 °C, indicating that this increase is very important for the acclimation of banana to LT stress.

In the present study, RNA-Seq techniques were employed to compare the changes in the expression of 42 *MaCsls* between the CT and CS banana cultivars subjected to LT conditions. The results revealed that these genes were differentially regulated by LT stress. For example, some genes (e.g., *MaCslC3* and *MaCslD1*) were upregulated by LTs in both cultivars, while others (e.g., *MaCslA5* and *MaCslC10*) were downregulated. Some genes showed different expression patterns under LT stress in the CS and CT cultivars. For example, *MaCslC5* in the CT cultivar showed lower expression levels under LT stress in comparison to 25 °C, but it showed opposite trend in the CS cultivar. On the other hand, all LTs induced the expression of *MaCslD4* in the CT cultivar, but this was not the case for the CS one. Similarly, the *CslD1* and *CslD4* levels of chilling-tolerant *indica* rice were upregulated by LTs, while the result was the opposite for *CslA1* and *CslF6* [46]. The phenomenon that genes from the same family differently respond to the same stress is frequently observed in the plant kingdom, such as *MaFLAs* (fasciclin-like AGP) in banana under LT stress [10] and barley *HvCslFs* upon the infection of cereal cyst nematodes [40]. These results suggested that members from certain gene families play diverse roles in plant tolerance/resistance to biotic and abiotic stresses.

In the present study, most *MaCsls* were suppressed by LTs in both CS and CT banana cultivars. Similarly, boron deficiency resulted in the downregulation of *CslB5* and several xyloglucan endotransglucosylase/hydrolase proteins (XTHs) in Arabidopsis roots [37,38]. In spinach (*Spinacia oleracea* L.), *CslE1* was found to be inhibited by salinity stress and a combination of salinity and cadmium stress [36]. Some *Csl* and *CesA* genes in Arabidopsis were also found to negatively modulate salt tolerance [73]. Besides abiotic stress, plant *Csl* genes were also reported to be suppressed by pathogens. For instance, the expression of several *CesA* genes (homologs of *Csl* genes) in rice was downregulated by rice tungro spherical virus at the early stage of infection [74], while the expression level of *HvCslF6* in barley decreased immediately after pathogen infection [40].

On the other hand, *MaCslD4* was induced by LT in the CT cultivar, and showed significantly higher expression levels in the CT cultivar under LTs when compared to the CS one, suggesting this gene plays an important role in the chilling tolerance of banana. Besides *MaCslD4*, *MaCslA4/12* and *MaCslE2* are also likely related to banana chilling tolerance, because they showed higher expression levels in the CT cultivar when compared to the CS, and were induced by LT or remained stable under LT stress in the CT cultivar. Similarly, the *AtCslG3* in Arabidopsis showed a higher expression level in the Cd-tolerant ecotype after exposure to Cd stress, likely being involved in enhanced Cd retention in the cell wall and reduction of Cd toxicity [39]. Another Arabidopsis *Csl* gene, *AtCslD5*, was suggested to play a critical role in osmotic stress, and is required for osmotic tolerance, because hypersensitive *sos6-1* (encodes *AtCslD5*) mutant plants accumulate high levels of reactive oxygen species under osmotic stress [35]. The silencing of *HvCslD2* resulted in the increased susceptibility of barley to powdery mildew, suggesting that *HvCslD2*-mediated cell wall changes represented an important defense reaction [41,75]. Thus, *Csl* genes are involved in plant tolerance/resistance to biotic and abiotic stresses [76]. The *MaCsl* genes found by this study could help the banana cell wall withstand the LT conditions. As proposed earlier, they might affect the cell turgor pressure [76], or increase cell wall flexibility/extensibility and reconstruction under LT-induced dehydration [72,75,76,77,78].

## 4. Materials and Methods

### 4.1. Plant Materials and Natural LT Conditions

The plant material for this study included two banana genotypes, *Musa* spp. AAA cv. Baxijiao and *Musa* spp. ABB cv. Dongguandajiao, which are CS and CT, respectively [9,79]. Three biological replicates of each genotype were subjected to low temperature treatments, following the method described by Yan et al. [9]. The leaves of plants growing for 3 days at 25 °C, 16 °C, 10 °C and 7 °C were used for analyses.

### 4.2. Measurement of Hemicellulose Content of Banana Leaves

The leaf samples were treated according to the analytical procedure recommended by the national renewable energy laboratory (NREL) [80] to obtain the filtrate. The content of hemicellulose in the filtrate was determined by the orcinol colorimetric method [81].

### 4.3. RNA-Seq Analysis

The RNA preparation, and the library preparation for RNA-Seq and data analysis, were carried out as described by Klepikova et al. [82]. False discovery rate was used to determine the threshold of the *p*-value in multiple tests and analyses. In the present study, |log2 (fold change)| > 1 and a threshold of false discovery rate values <0.05 were used as the threshold to evaluate the significance of differentially expressed genes.

### 4.4. Identification of Csls in Banana

To study the *Csl* gene family in banana, all *Musa acuminata* protein sequences were obtained from Banana-Genome-Hub (https://banana-genome-hub.southgreen.fr/download) *Musa acuminata* DH Pahang v2 (updated in January 2016). The Csl amino acid sequences of *Arabidopsis* and *Oryza sativa* were downloaded from the Arabidopsis Information Resource (TAIR) (http://www.Arabidopsis.org/download) and the Rice Genome Annotation Project (http://rice.plantbiology.msu.edu/downloads_gad.shtml).

Double-directional BLAST was employed to obtain potential Csl members; the BLAST function of TBTools [83] was used to retrieve potential MaCsl sequences referring to the amino acid sequence of Csl proteins of *Arabidopsis*. The obtained potential MaCsl sequences were thereafter compared with Csls in the Swissprot database, and those without typical characteristics of Csl proteins were removed. Candidate genes were obtained by analyzing the obtained genes using the Search pfam (http://pfam.xfam.org/search) online tool and eliminating sequences that lack the typical functional domain of Csls. All candidate MaCsls should contain one of the two PFAM domain models, namely PF00535 or PF03552.

### 4.5. Physicochemical Properties and Phylogenetic Analysis

Expert protein analysis system (ExPASy, http://web.expasy.org/compute_pi/) were employed to predict the pI and MW of MaCsl amino acid sequences.

Amino acid sequence alignments of all Csl members from the Arabidopsis, rice, or banana were performed using Clustal W2 [84] under default settings, while Molecular Evolutionary Genetics Analysis (MEGA) 7.0 [85] software was used to construct the phylogenetic trees, followed by visualization with FigTree v1.4.2 (http://tree.bio.ed.ac.uk/software/figtree/). The default parameters were manually adjusted using the neighbor-joining method (the JTT+I+G substitution model and 1000 bootstrap replicates).

### 4.6. Conserved Motif and Gene Structure Analysis

The online software MEME (http://meme-suite.org/tools/meme) was used to identify the conserved motifs of MaCsl from the deduced protein sequence. The number of conserved motifs selected was 10, while the other default parameters were set automatically by the software. The results were visualized with TBtools. The structural analysis of the introns, exons and non-coding regions of all MaCsls was performed by Gene Structure Display Server 2.0 (GSDS, https://gsds.cbi.pku.edu.cn/) [86] using corresponding CDS sequences and genomic sequences of MaCsls retrieved from Banana-Genome-Hub (https://banana-genome-hub.southgreen.fr/download).

### 4.7. Quantification of the Expression Level of MaCsls Using qPCR

The experiment was carried out according to the method described by Meng et al. [10] and the primers used are listed in Appendix A.

### 4.8. Statistical Analysis

One-way analysis of variance (one-way ANOVA) was done using IBM SPSS Statistics software for Windows, Version 26.0 (IBM Corporation, Armonk, NY, USA). The results of hemicellulose content and qPCR were expressed as mean ± SE. Statistical differences between the two species at each temperature point were determined using the Student’s *t*-test.

## 5. Conclusions

This study provides a comprehensive analysis of the *MaCsl* gene family in banana. In total, 42 members of the *MaCsl* gene family were identified in the *Musa acuminata* genome, and were classified into six subfamilies: *CslA*, *CslC*, *CslD*, *CslE*, *CslG*, and *CslH*. This information provides an important basis for studying the physiological role played by the *MaCsl* genes in response to biotic and abiotic stresses. In addition, the different expression patterns of *MaCsl* genes in CT and CS banana cultivars under LT stress indicate their involvement in plant chilling tolerance, suggesting their potential utilization in the breeding of CT banana. Further research should focus on the function of specific *MaCsl* genes in chilling tolerance and the underlying mechanisms.

## Figures and Tables

**Figure 1 plants-10-00122-f001:**
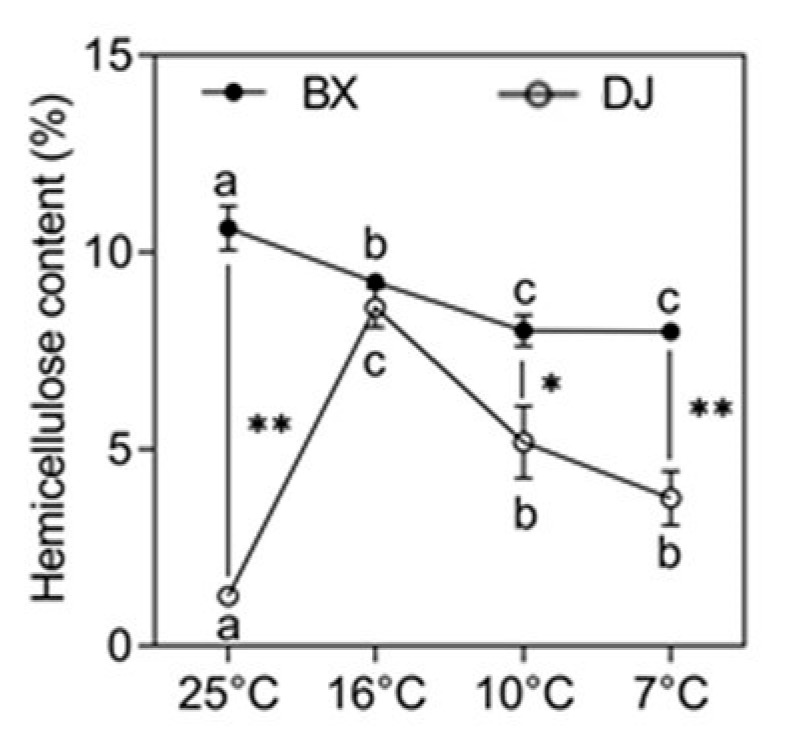
Changes in the hemicellulose content of banana (*Musa* spp.) under low temperature stress. Hemicellulose content is expressed as a percentage per gram of fresh leaves. The data represent an average of three replicates ± SE. Values followed by the same letter are not significantly different using a Duncan’s multiple range test at *p* < 0.05 after angular transformation of the data for each cultivar. Values marked with a star were considered significant at *p* < 0.05, while values marked with two stars were considered significant at *p* < 0.01 when evaluated using Student’s *t*-test. BX ‘Baxijiao’, chilling-sensitive (CS); DJ ‘Dongguandajiao’, chilling-tolerant (CT).

**Figure 2 plants-10-00122-f002:**
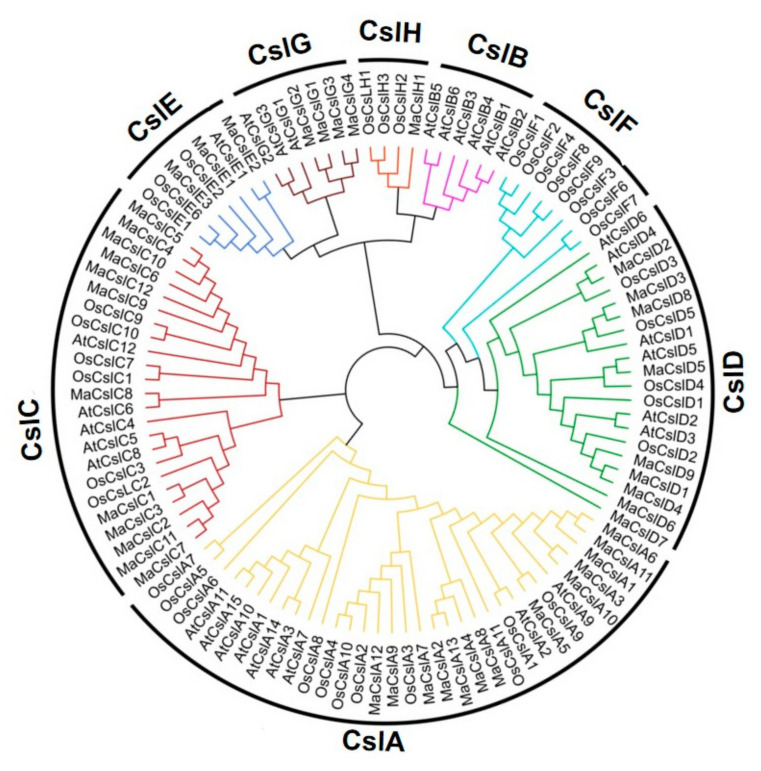
Phylogenetic tree of Csl proteins in banana (*Musa acuminata*), rice (*Oryza sativa*), and Arabidopsis (*Arabidopsis thaliana*). Totals 42 Csl proteins in banana, 36 in rice, and 30 in Arabidopsis were analyzed using Clustal W. Neighbor-joining trees were constructed using MEGA7.0. The bootstrap value was 1000 replicates.

**Figure 3 plants-10-00122-f003:**
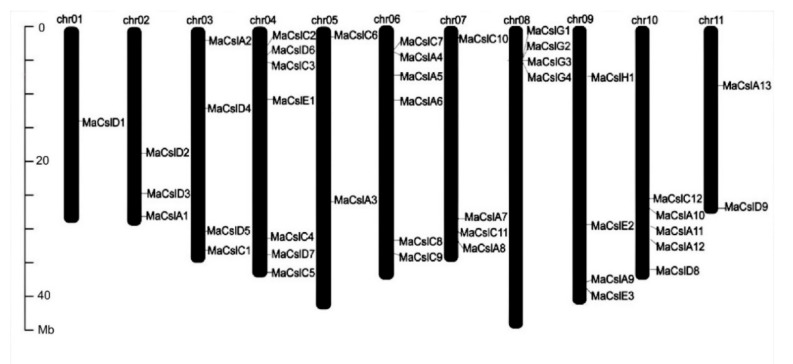
Chromosomal localization of the *MaCsl* genes in the banana (*Musa acuminata*) genome.

**Figure 4 plants-10-00122-f004:**
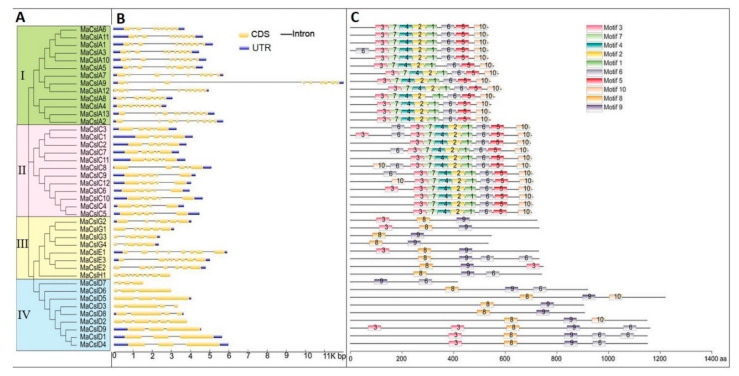
Phylogenetic relationships, gene structure, and motif distribution of MaCsls. (**A**) Unrooted phylogenetic tree of 42 MaCsls, generated with the MEGA7.0 software by the neighbor-joining method with 1000 bootstrap replicates after alignment of the full-length protein sequence by Clustal W. (**B**) Exon/intron structures of *MaCsl* genes. (**C**) Conserved motifs of the *MaCsl* gene family, analyzed by Multiple Em for Motif Elucidation (MEME). Different motifs are represented by different colored boxes with numbers 1–10 (color figure online).

**Figure 5 plants-10-00122-f005:**
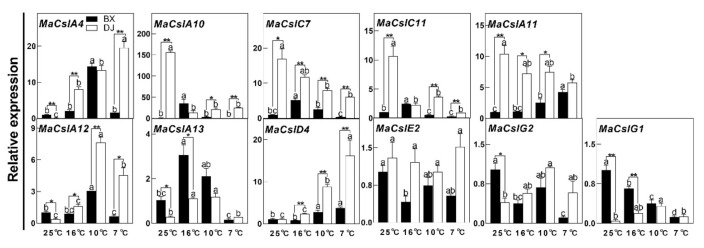
Quantitative real-time PCR (qPCR) analysis of 11 *MaCsl* gene expressions in banana (*Musa* spp.) leaves under low temperature stress. The data represent an average of three replicates ± SE. Values followed by the same letter are not significantly different using Duncan’s multiple range test at *p* < 0.05 after the angular transformation of the data for each cultivar (comparison among different low-temperature points). Values marked with a star were considered significant at *p* < 0.05, while values marked with two stars were considered significant at *p* < 0.01 when evaluated using Student’s *t*-test (comparison between the two cultivars at the same low-temperature point). BX ‘Baxijiao’, chilling-sensitive; DJ ‘Dongguandajiao’, chilling-tolerant.

**Table 1 plants-10-00122-t001:** 42 *MaCsl* genes identified in *Musa acuminata* and their sequence characteristics.

Gene Name	Gene ID	Chr	Start	End	Length (bp)	Strand	Size (aa)	pI	MW (kD)
*MaCslA1*	*Ma02_t22150*	chr02	27,565.079	27,570.275	1602	−	534	8.9	60,834.3
*MaCslA2*	*Ma03_t01730*	chr03	1207.899	1213.633	1629	+	543	9.0	61,799.5
*MaCslA3*	*Ma05_t18900*	chr05	25,534.488	25,538.961	1602	−	534	9.0	60,875.2
*MaCslA4*	*Ma06_t04300*	chr06	3100.141	3102.901	1635	−	545	8.7	62,234.1
*MaCslA5*	*Ma06_t09180*	chr06	6479.551	6484.223	1668	+	556	9.0	63,946.1
*MaCslA6*	*Ma06_t14920*	chr06	10,164.997	10,168.70	1602	+	534	8.9	60,781.3
*MaCslA7*	*Ma07_t19410*	chr07	27,422.775	27,428.514	1719	+	573	9.1	65,060.1
*MaCslA8*	*Ma07_t22600*	chr07	30,481.370	30,484.454	1677	+	559	9.0	64,175.8
*MaCslA9*	*Ma09_t27610*	chr09	38,562.778	38,574.772	1626	+	542	8.8	61,639.2
*MaCslA10*	*Ma10_t11450*	chr10	24,986.818	24,991.666	1602	+	534	8.9	61,025.5
*MaCslA11*	*Ma10_t15510*	chr10	27,582.419	27,587.095	1602	−	534	9.1	60,994.7
*MaCslA12*	*Ma10_t18740*	chr10	29,549.996	29,554.980	1755	−	585	8.9	65,936.2
*MaCslA13*	*Ma11_t08690*	chr11	6915.612	6920.893	1626	+	542	9.3	62,165.2
*MaCslC1*	*Ma03_t29290*	chr03	32,234.549	32,238.695	2106	+	702	8.9	79,991.8
*MaCslC2*	*Ma04_t02130*	chr04	1873.146	1876.963	2094	−	698	8.7	79,922.4
*MaCslC3*	*Ma04_t05930*	chr04	4437.079	4440.382	2085	+	695	9.1	79,398.2
*MaCslC4*	*Ma04_t29650*	chr04	30,511.868	30,515.518	2115	+	705	8.5	79,553.3
*MaCslC5*	*Ma04_t38760*	chr04	36,164.056	36,168.517	2106	+	702	7.8	79,537.2
*MaCslC6*	*Ma05_t01870*	chr05	1142.347	1146.294	2112	−	704	8.5	79,580.2
*MaCslC7*	*Ma06_t03600*	chr06	2622.347	2625.771	2067	+	689	8.8	79,142.5
*MaCslC8*	*Ma06_t29550*	chr06	30,901.418	30,906.541	2091	+	697	9.1	78,791.9
*MaCslC9*	*Ma06_t31890*	chr06	32,901.867	32,906.161	2115	+	705	8.1	79,750.6
*MaCslC10*	*Ma07_t00740*	chr07	619,565	624,194	2121	−	707	8.0	79,935.6
*MaCslC11*	*Ma07_t20970*	chr07	28,958.080	28,961.838	2106	−	702	8.8	80,597.2
*MaCslC12*	*Ma10_t09350*	chr10	23,545.265	23,549.303	2124	−	708	7.5	79,820.2
*MaCslD1*	*Ma01_t18500*	chr01	13,756.260	13,761.905	3450	+	1150	6.9	128,482.2
*MaCslD2*	*Ma02_t07580*	chr02	18,211.558	18,215.380	3447	−	1149	6.4	128,432.9
*MaCslD3*	*Ma02_t17080*	chr02	24,143.640	24,146.984	2709	−	903	8.9	100,568.6
*MaCslD4*	*Ma03_t14070*	chr03	11,229.456	11,235.431	3453	+	1151	7.5	128,361.1
*MaCslD5*	*Ma03_t25420*	chr03	29,461.721	29,465.754	3657	−	1219	8.1	134,887.3
*MaCslD6*	*Ma04_t04560*	chr04	3486.824	3489.813	2757	+	919	8.8	103,194.6
*MaCslD7*	*Ma04_t33100*	chr04	32,908.316	32,909.842	1245	−	415	9.5	46,143.7
*MaCslD8*	*Ma10_t26210*	chr10	34,009.560	34,013.193	2721	−	907	8.9	100,850.9
*MaCslD9*	*Ma11_t21750*	chr11	25,761.358	25,765.90	3480	+	1160	6.8	128,916.7
*MaCslE1*	*Ma04_t13090*	chr04	9902.007	9907.918	2187	−	729	8.3	83,277.4
*MaCslE2*	*Ma09_t20060*	chr09	27,488.282	27,493.065	2241	+	747	8.2	84,590.1
*MaCslE3*	*Ma09_t28670*	chr09	39,320.301	39,325.310	2193	+	731	8.5	82,679.5
*MaCslG1*	*Ma08_t05160*	chr08	3537.407	3540.538	2193	−	731	8.2	81,197.7
*MaCslG2*	*Ma08_t05170*	chr08	3544.521	3548.562	2169	−	723	6.9	80,645.9
*MaCslG3*	*Ma08_t05180*	chr08	3548.894	3551.285	1638	−	546	6.7	60,810.7
*MaCslG4*	*Ma08_t05190*	chr08	3561.425	3563.941	1602	−	534	6.5	59,472.2
*MaCslH1*	*Ma09_t08420*	chr09	5566.954	5569.888	2223	+	741	7.2	83,277.7

aa: amino acids; bp: base pair; kD: kilodaltons; MW: molecular weight; pI: isoelectric point.

**Table 2 plants-10-00122-t002:** Analysis of differentially expressed *MaCsl* genes in banana (*Musa* spp.) leaves under low temperatures.

Gene Name	log2 Fold Change									
	CKDJ vs. CKBX	LT16DJ vs. LT16BX	LT10DJ vs. LT10BX	LT7DJ vs. LT7BX	LT16BX vs. CKBX	LT10BX vs. CKBX	LT7BX vs. CKBX.	LT16DJ vs. CKDJ	LT10DJ vs. CKDJ	LT7DJ vs. CKDJ
*MaCslA1*								−2.52		
*MaCslA2*										
*MaCslA3*								−1.73		−2.44
*MaCslA4*	−2.10							1.51	2.00	2.31
*MaCslA5*					−1.70	−1.99	−2.20	−2.46	−2.88	−2.81
*MaCslA6*										
*MaCslA7*							−2.63		−1.18	−2.20
*MaCslA8*	2.31			−1.82			3.98			
*MaCslA9*			−1.35					−1.42	−1.41	−1.44
*MaCslA10*	3.11			3.94				−3.60	−4.16	−3.94
*MaCslA11*			1.96	2.12				−1.76	−2.17	−2.20
*MaCslA12*		−1.59					−1.73		1.14	
*MaCslA13*				1.85			−3.81			−1.71
*MaCslC1*										
*MaCslC2*						−3.54	−6.28	−4.44	−5.34	−5.66
*MaCslC3*							4.90			3.93
*MaCslC4*										
*MaCslC5*	2.96		−3.23			3.17	3.75	−2.28	−3.07	−1.95
*MaCslC6*								−1.90	−1.65	
*MaCslC7*	3.86		2.59	3.37				−2.97	−3.22	−3.20
*MaCslC8*						−1.05	−1.22			−1.10
*MaCslC9*									−1.61	
*MaCslC10*						−3.45	−3.96	−4.22	−3.94	−3.53
*MaCslC11*	1.68		1.94			−3.00	−3.78	−2.85	−3.27	−4.51
*MaCslC12*									−1.26	
*MaCslD1*							1.58			1.34
*MaCslD2*							4.04			
*MaCslD3*										
*MaCslD4*			1.27	1.34					1.52	2.16
*MaCslD5*							2.46			2.70
*MaCslD6*										
*MaCslD7*										
*MaCslD8*										
*MaCslD9*	−1.28						3.71	1.89		4.41
*MaCslE1*							−1.27			
*MaCslE2*	1.50	1.98	1.37	2.58	−1.55	−1.70	−2.11	−1.16	−1.89	
*MaCslE3*			−1.01				−1.97		−1.16	
*MaCslG1*		1.33	1.30	2.74			−2.31			
*MaCslG2*	−1.17			1.79		−1.03	−4.03			
*MaCslG3*	−1.54	−1.41				−3.18				
*MaCslG4*										
*MaCslH1*	−3.09	−1.38			−2.41	−4.62				−3.46

BX: ‘Baxijiao’, chilling-sensitive; DJ: ‘Dongguandajiao’, chilling-tolerant; CK: 25 °C control; LT16: low temperature of 16 °C; LT10: low temperature of 10 °C; LT7: low temperature of 7 °C.

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
