# Peer review of "Genome-Wide Identification of Banana Csl Gene Family and Their Different Responses to Low Temperature between Chilling-Sensitive and Tolerant Cultivars"

_plants, 2021, doi:10.3390/plants10010122_

Round 1
Reviewer 1 Report
In this manuscript, the authors provide a new insight on the genes involved the complex pathways for cold stress tolerance in banana. Given the drastic changes in climatic conditions, this information will be relevant when breeding bananas for subtropical conditions where productivity can be affected by cold stress.
Overall, the work is well presented but I have some comments and edits that I believe will improve the manuscript when addressed. Details are provided in the attached file.

Reviewer 2 Report
This is a very interesting paper, revealing new data about Csl protein family starting from in silico research and tested by molecular screening on several low temperature stresses for banana plant. I found these research and data solid with enough merit to be published but it is necessary some attention on writing as some phrases seems to be truncated. Discussion is directed to Csl genes and well documented with references.
My suggestions:
line 93- ...expression of 42 MaCsl members... instead of ...expression of the all 42 MaCsl members...
line 181-2.3. Differences in the responses of MaCsls to LT stress between CS and CT cultivars - here I recommend the presentation of a more complete description of RNA-Seq data (total number of expressed and DEG genes for each stress condition and respective samples).
line 210 - in the figure 5 description black bar and white bar are not described. is white/black for CS/CT or BX/DJ?
line 300- ...Cd retention in CW and reduction of Cd toxicity - define CW (Cell wal?)
line 321- second period of this paragraph seems incomplete or truncated after [75] reference. It must be re-writted.
Reviewer 3 Report
The manuscript from Yuan et al. advance our knowledge in the characterization of mechanisms that confer chilling tolerance in banana. The techniques used are appropriate and the results are well presented, however some modifications are needed for publication:
- In the introduction the authors must better clarify why they speculate/investigate a role for the hemicellulose and Csl genes in chilling tolerance (Line 76). Is there any report for other species in the literature?
- Line 95, the abbreviation and the description of the CS and CS cultivar must be explain in the main text.
- Line 134 the sentence “In addition, most MaCslA 134 and MaCslC genes possess both upstream and downstream.” Is not clear, please reformulate.
- Line 153, the paragraph on the phylogenetic analysis must be placed earlier in the text, as first paragraph in the result section 2.2. In paragraph 2.2.1 line 111 the authors mention that the MaCls genes were identified based on phylogenetic relationship with rice and arabidopsis.
- Table 2. In the table only the Log2FC is reported, the authors must add the p-value (or other statistics) either in the main table or in the supplements for the reader to assess the significance of the observed change in expression.
- Line 191. The paragraph on the validation of the RNA-seq results by qPCR should be improved. It is not clear why the authors chose to validate genes that were not found differentially expressed between the two cultivars in the RNA-seq, such as MaClsA4, MaClsA7, MaClsC4 or MaClsD8. A substantial number of genes (MaClsA12, MaClsA13, MaClsD9, MaClsE2, MaClsG1, MaClsG3) the RNA-seq and qPCR results are not consistent, this is not mentioned in the text. Finally, the authors conclude that “MaCslA4/7/12, MaCslC4, MaCslD4/8, MaCslE2 were more likely involved in the chilling tolerance of banana to LT” (line 193). This conclusion is not fully supported by the expression data because some of the genes mentioned were not differentially expressed between the two cultivars in the RNA-seq (MaClsA4, MaClsA7 and MaClsC4) or could not be confirmed in the qPCR analysis (MaClsA12 and MaClsE2).
- Figure 5. In the legend the Authors must specify which cultivar is represented in black and which one in white in the histograms in the figure.
- Line 231. “Banana only contain one CslH gene, and lack of CslF and CslJ genes, suggesting that the much lower abundance/level of (1, 3; 1, 4)-β-glucan in banana than in the other monocotyledon crops such as rice, wheat and maize”. Can the authors speculate on a biological reason for the lower abundance?
- Additional point for the discussion. RNA-seq experiment upon temperature treatment have been performed in other species (Arabidopsis for example). Where Csl genes found differentially expressed? Or is this response specific only for banana?
Reviewer 4 Report
Yuan and colleagues characterized the cellulose synthase-like (Csl) genes in banana, responsible for hemicellulose synthesis. There is merit in the work. Probably this is the first characterization and classification of Csl genes in banana, including sequencing, phylogenetic analysis and chromosomal localization as well as motif composition of the encoded proteins. Interestingly, they even found members of the seemingly dicot-specific CslG family.
However I’m not that convinced about the other part of the study, the proposed involvement of Csl genes in low temperature tolerance. Based on literature data on other osmotic stresses, the Authors argue that Csl genes might affect the cell turgor pressure, or increase wall flexibility/extensibility and reconstruction under LT-induced dehydration. However the chilling tolerant variety had the lower hemicellulose content in the study. As result of cold treatment, hemicellulose content of one of the tested banana cultivars decreased, while that of the other has increased, then decreased. The authors theorize that it was the amount of certain types of hemicellulose instead of the total hemicellulose content affecting the chilling tolerance of banana. Some measurements of hemicellulose types could have been useful at this step, to confirm their idea. Furthermore, expression of some of the Csl genes has increased and the expression of others has decreased during the coldness. Enzyme kinetic characterisation of the upregulated/downregulated Csl enzymes could have also helped to understand the observed changes.
Minor issues:
The Csl genes were uncovered using transcriptome sequencing. Did it yield other cold-stress responsive genes as well?
Round 2
Reviewer 4 Report
The manuscript has improved a lot. While I did not get the further measurements which I required, I understand that it is mostly a sequencing and phylogenetic groundwork study, as the Authors also write that starting from their results „Further research should focus on the function of specific MaCsl genes in chilling tolerance and the underlying mechanisms”. I also see that the other reviewers were less critical about the results in general. If they let it go, I do not object either, the MS can be published.